# IMPROVING POLICY OPTIMIZATION VIA ENHANCED EXPLORATION

## ABSTRACT

Reinforcement learning has become the standard approach for aligning large language models to complex reasoning tasks. However, these methods often overlook rare valuable responses, as learning signals are dominated by high-probability, frequently sampled outputs. To address this, we propose **EX**ploration-Enhanced **P**olicy **O**ptimization (EXPO), a novel approach that dynamically reweights the advantage of each response based on its generation probability. EXPO amplifies gradients from rare valuable samples, ensuring they contribute meaningfully to policy updates and guide the model toward underexplored, high-value solutions. We evaluate EXPO on multiple mathematical reasoning benchmarks. It consistently outperforms strong baselines across model scales: on Qwen2.5-Math-1.5B, EXPO surpasses DAPO by +3.0%; on Llama-3.2-3B-Instruct, by +3.6%; and on the larger Qwen2.5-Math-7B, it outperforms the DAPO by +4.6%, Dr.GRPO by +5.3% and instruction-tuned baseline by +9.1%, These gains demonstrate EXPO's effectiveness in leveraging valuable but underrepresented responses for better policy learning.

## 1 INTRODUCTION

The development of reasoning-centric LLMs, including OpenAI-o3 (OpenAI, 2025), DeepSeek-R1 (DeepSeek-AI et al., 2025), and Kimi-K2 (Bai et al., 2025), has significantly advanced the frontier of LLM capabilities, particularly in tackling complex reasoning tasks in mathematics and programming (Yue et al., 2025). This progress is primarily driven by Reinforcement Learning with Verifiable Rewards (RLVR). Most implementations rely on policy gradient algorithms, with Proximal Policy Optimization (PPO) (Schulman et al., 2017) and its recent variants (Fan et al., 2025; Ren et al., 2025; Cozma et al., 2025) being widely adopted for their stability and empirical effectiveness. These methods iteratively refine the policy by estimating the advantage of sampled responses and reinforcing high-reward behaviors.

Despite their promising results, these methods remain fundamentally constrained by the base LLM's initial capabilities (Yan et al., 2025;

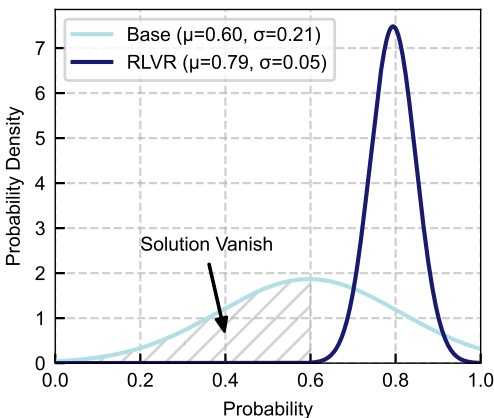

Figure 1: **Probability Density of Response Before and After RLVR Training.** RLVR sharpens the distribution and eliminates low-probability responses, reducing diversity and suppressing rare valuable reasoning paths.

Zhao et al., 2025). Reinforcement learning amplifies existing behaviors by biasing the policy toward high-reward paths, which improves sampling efficiency . However, this gain comes at a cost: it narrows the model's reasoning scope(Yue et al., 2025). We observe that this limitation stems from the suppression of low-probability trajectories, even when they produce exceptionally high rewards. As illustrated in Figure 2, this suppression causes a collapse in response diversity and weakens the model's ability to solve problems creatively. We analyze the root cause lies in the expectation-based

nature of the policy gradient objective, which weights gradient updates by the likelihood of sampled trajectories. As a result, high-probability actions dominate learning, while rare valuable responses contribute negligibly to the gradient.

To overcome this limitation, we propose that the policy should be explicitly incentivized to learn from its valuable and often ignored discoveries. Amplifying the learning signal from these rare valuable samples is critical not only for escaping local optima but also for unlocking the model's full creative and problem-solving capacity. Motivated by this insight, we introduce **Exploration-Enhanced Policy Optimization (EXPO)**, a novel algorithm that dynamically reshapes the training objective to prioritize high-reward, low-probability responses.

EXPO achieves this by introducing a dynamic weighting mechanism that modulates the advantage of each sampled response based on its likelihood under the current policy. Specifically, for desirable (high-reward) responses, the weight is inversely proportional to their generation probability, thereby amplifying gradients for rare valuable outputs. Conversely, for undesirable responses, the weight places stronger emphasis on penalizing frequent mistakes, encouraging the policy to avoid harmful or suboptimal behaviors. We summarize our contributions as follows:

- We identify and analyze a fundamental limitation of standard policy gradient methods such as GRPO and its variants in aligning large language models: their tendency to overlook low-probability, high-reward responses, which hinders effective exploration.
- We propose EXPO, a novel and lightweight algorithm that mitigates this bias by dynamically reweighting advantages based on response probability, thereby focusing learning on the most informative samples.
- We demonstrate through extensive experiments on mutiple mathematical tasks that EXPO consistently outperforms strong baselines, yielding models that better explore the reward landscape and generate higher-quality, more diverse outputs.

## 2 PRELIMINARIES

We build on recent advances in policy gradient methods for LLM post-training to improve performance. Group Relative Policy Optimization (GRPO) (Shao et al., 2024b) replaces PPO's value network with group-based advantage normalization. For a prompt $q$, it samples $G$ responses $\{o_1, \ldots, o_G\}$, computes rewards $\{R_1, \ldots, R_G\}$, and normalizes advantages within the group:

$$A_{i,t} = \frac{R_i - \text{mean}(\{R_i\}_{i=1}^G)}{\text{std}(\{R_i\}_{i=1}^G) + \epsilon}. \tag{1}$$

where $\epsilon$ ensures numerical stability. The GRPO objective includes KL regularization:

$$\mathcal{J}_{\text{GRPO}}(\theta) = \mathbb{E}\left[\frac{1}{G}\sum_{i=1}^G \frac{1}{|o_i|}\sum_{t=1}^{|o_i|}\left(\min\left(r_{i,t}A_{i,t}, \text{clip}(r_{i,t}, 1-\varepsilon, 1+\varepsilon)A_{i,t}\right) - \beta\mathbb{D}_{\text{KL}}(\pi_\theta\|\pi_{\text{ref}})\right)\right]. \tag{2}$$

with $r_{i,t} = \pi_\theta(o_{i,t} \mid q, o_{i<t})/\pi_{\theta_{\text{old}}}(o_{i,t} \mid q, o_{i<t})$ and $\beta$ controlling KL penalty strength. DAPO (Yu et al., 2025) extends GRPO with four key improvements: (1) asymmetric clipping bounds $(\varepsilon_{\text{low}}, \varepsilon_{\text{high}})$ for more flexible updates, (2) dynamic sampling to adjust group composition, (3) token-level loss for finer control, and (4) reward shaping for long responses. Its objective removes the per-response normalization and KL term, instead applying a diversity constraint:

$$\mathcal{J}_{\text{DAPO}}(\theta) = \mathbb{E}\left[\frac{1}{\sum_i |o_i|}\sum_{i=1}^G\sum_{t=1}^{|o_i|}\min\left(r_{i,t}A_{i,t}, \text{clip}(r_{i,t}, 1-\varepsilon_{\text{low}}, 1+\varepsilon_{\text{high}})A_{i,t}\right)\right], \tag{3}$$

$$\text{s.t.} \quad 0 < |\{o_i \mid \texttt{is\_equivalent}(o_i, a)\}| < G.$$

which ensures sampled responses are not all identical to the reference answer $a$, preserving output diversity. These methods form the foundation for our approach, which further addresses their shared limitation: the suppression of rare valuable responses due to expectation-based gradient weighting.

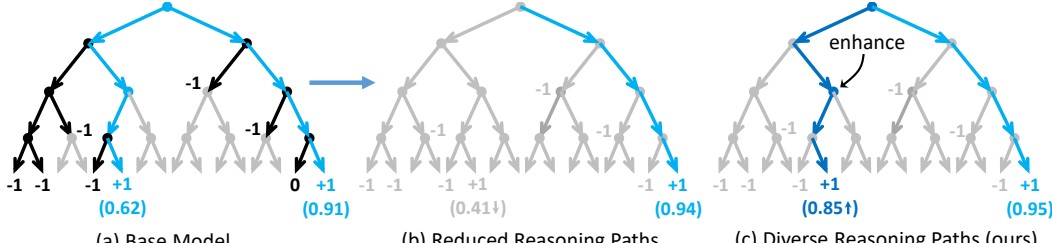

Figure 2: **Illustration of Reasoning Path Dynamics During Policy Optimization.** Solutions are scored +1 if correct and -1 otherwise; numbers in () indicate their respective probabilities. We observe: (a) Before optimization, the base model explores a diverse set of reasoning paths. (b) After standard reinforcement learning optimization, the reasoning space collapses: only high-reward, high-probability paths remain dominant, at the expense of diversity. (c) Our method preserves and enhances exploration by dynamically amplifying learning signals from rare but high-reward paths, achieving both improved performance and sustained reasoning diversity.

## 3   MOTIVATION

RLVR for large language models typically begins by sampling a large batch of responses, which is then partitioned into smaller mini-batches for iterative training. Responses that align with desired behavior receive higher rewards, encouraging the model to reinforce them; conversely, poorly aligned responses are penalized, prompting the model to suppress them. In essence, RLVR amplifies existing positive behaviors, but this comes at a cost.

As illustrated in Figure 1, we observe that after RLVR training, the model's output distribution becomes more concentrated: response probabilities increase on average, and their variance decreases. Many low-probability responses that existed in the base model, including numerous correct or high-quality ones, vanish entirely. This phenomenon is often attributed to *entropy collapse*, where the policy distribution narrows over time, leading to a significant loss in solution diversity.

We offer a complementary, yet often overlooked, perspective. Since RLVR training relies on sampling responses from the current policy before updating, high-probability responses are sampled more frequently and thus dominate gradient updates. In contrast, low-probability responses, even when they yield high rewards, are rarely sampled and exert minimal influence on learning. This imbalance intensifies over time: as the model becomes more confident in its frequent outputs, those outputs increasingly steer future updates, creating a self-reinforcing loop that suppresses diversity and encourages homogeneity.

This intuition is formally validated by examining the policy gradient objective. The core issue lies in its expectation-based formulation:

$$\nabla_\theta J(\theta) = \mathbb{E}_{\tau \sim \pi_{\theta_{old}}} \left[ G(\tau) \nabla_\theta \log \pi_\theta(\tau) \right] = \sum_\tau \pi_{\theta_{old}}(\tau) \cdot G(\tau) \nabla_\theta \log \pi_\theta(\tau). \quad (4)$$

where $G(\tau)$ denotes the advantage of trajectory $\tau$. Because each term is weighted by the sampling probability $\pi_{\theta_{old}}(\tau)$, even exceptionally rewarding responses contribute negligibly to the gradient if they are unlikely under the current policy.

This leads to what we term **statistical short-sightedness**: rare but excellent responses, often happy accidents, are drowned out by the statistical mass of common, mediocre ones. The model thus becomes proficient at refining what it already knows, while systematically ignoring its flashes of brilliance. The result is premature convergence: models that generate safe, predictable outputs, but fail to discover superior, creative solutions lurking in the long tail of the response space. This motivates our proposal: to move beyond the standard expectation and introduce a mechanism that **dynamically amplifies the learning signal from high-reward, low-probability discoveries**, compelling the model to attend to and learn from its promising and low-probability outputs.

## 4 METHOD

### 4.1 EXPLORATION-ENHANCED POLICY OPTIMIZATION

To address the suppression of rare valuable responses in policy gradient methods, we introduce Exploration-Enhanced Policy Optimization, a novel algorithm that dynamically reweights the learning signal based on a response's probability. EXPO amplifies gradients from outputs that are high-reward but low-probability under the current policy, ensuring they contribute meaningfully to policy optimization. The core mechanism modifies the advantage $A_i$ of each response $y_i$ using a sequence-level **dynamic weight** $\alpha_i$:

$$\hat{A}_i = (1 + \alpha_i)A_i. \tag{5}$$

where $A_i$ is the standard group-normalized advantage. The weight $\alpha_i$ is large for rare valuable responses and near zero otherwise. We define $\alpha_i$ as:

$$\alpha_i = \text{clip}\left((1 - \text{clip}(\tilde{p}_i, \delta, 1))^\gamma, 0, \alpha_{\max}\right). \tag{6}$$

with the effective probability $\tilde{p}_i$ set to: $p_i = \pi_{\theta_{\text{old}}}(y_i \mid x)$ if $A_i > 0$ (amplify rare valuable responses), $1 - p_i$ if $A_i < 0$ (penalize frequent mistakes). Here, $\gamma \geq 0$ controls the focus on rare responses. The value of the advantage $A_i$ is very sensitive to model training, to ensure stable training, we introduce two mechanisms:

- **Progressive Adjust** ($\delta$): We set $\delta = t/T$, as training progresses ($\delta \to 1$), amplification of rare responses fades smoothly, preventing late-stage instability.
- **Weight Clamping** ($\alpha_{\max}$): We set an upper limit on $\alpha_i \leq \alpha_{\max}$ (e.g., 0.5), bounding the scaling factor $(1 + \alpha_i) \in [1.0, 1.5]$ to avoid extreme updates.

Substituting the reweighted advantage $\hat{A}_i$ into the DAPO objective yields the EXPO objective.

$$\mathcal{J}_{\text{EXPO}}(\theta) = \mathbb{E}_{(q,a) \sim \mathcal{D}, \{o_i\}_{i=1}^G \sim \pi_{\theta_{\text{old}}}(\cdot|q)}$$

$$\left[ \frac{1}{\sum_{i=1}^G |o_i|} \sum_{i=1}^G \sum_{t=1}^{|o_i|} \min\left( r_{i,t}(\theta)\hat{A}_{i,t}, \text{clip}\left(r_{i,t}(\theta), 1 - \varepsilon_{\text{low}}, 1 + \varepsilon_{\text{high}}\right)\hat{A}_{i,t} \right) \right]. \tag{7}$$

This formulation ensures that rare valuable responses have a much stronger effect on the policy update, enabling EXPO to learn from exceptional but infrequent outputs that standard methods overlook.

### 4.2 GRADIENT ANALYSIS

To better understand how EXPO reshapes the learning signal, we analyze its objective gradient. For clarity, we omit the PPO-style ratio clipping (i.e., assume clipping bounds are not active), and recall that both the advantage $A_i$ and the dynamic weight $\alpha_i$ are computed using the frozen old policy $\pi_{\theta_{\text{old}}}$ and are thus treated as constants during gradient computation.

Under these conditions, the gradient of the EXPO objective (Eq. 7) with respect to $\theta$ is:

$$\nabla_\theta \mathcal{J}_{\text{EXPO}}(\theta) = \mathbb{E}_{(q,a) \sim \mathcal{D}, \{o_i\}_{i=1}^G \sim \pi_{\theta_{\text{old}}}(\cdot|q)} \left[ \frac{1}{\sum_{i=1}^G |o_i|} \sum_{i=1}^G (1 + \alpha_i)A_i \sum_{t=1}^{|o_i|} \nabla_\theta \log \pi_\theta(o_{i,t} \mid q, o_{i<t}) \right]. \tag{8}$$

where we use the identity $\nabla_\theta r_{i,t}(\theta) = r_{i,t}(\theta)\nabla_\theta \log \pi_\theta(o_{i,t} \mid q, o_{i<t})$. In contrast, the gradient of the standard DAPO objective (Eq. 3) is:

$$\nabla_\theta \mathcal{J}_{\text{DAPO}}(\theta) = \mathbb{E}_{(q,a) \sim \mathcal{D}, \{o_i\}_{i=1}^G \sim \pi_{\theta_{\text{old}}}(\cdot|q)} \left[ \frac{1}{\sum_{i=1}^G |o_i|} \sum_{i=1}^G A_i \sum_{t=1}^{|o_i|} \nabla_\theta \log \pi_\theta(o_{i,t} \mid q, o_{i<t}) \right]. \tag{9}$$

The only difference between the two gradients is the multiplicative factor $(1 + \alpha_i)$ in EXPO. For high-reward responses ($A_i > 0$), $\alpha_i$ decreases with probability, it amplifies the influence of rare valuable outputs while reducing the impact of common ones. For low-reward responses ($A_i < 0$), $\alpha_i$ increases with probability, which discourages the model from assigning high likelihood to frequent mistakes. **This ensures that exploration is guided by quality, not just novelty**.

## 5 EXPERIMENTS

### 5.1 SETUP

**Datasets.** For training, we use the MATH dataset (Hendrycks et al., 2021), focusing on problems from difficulty levels 3 to 5. As shown in Figure 3, we further group these problems into five tiers based on the base model's performance, with the *Hard* tier exhibiting the most effective training dynamics, our experiments use **1,000** samples for training from this tier, see Appendix A for details. For evaluation, we benchmark our method across five mathematical reasoning datasets: (1) **AIME24**: A collection of 30 high-school olympiad-level problems from the 2024 American Invitational Mathematics Examination (Li et al., 2024). (2) **AMC**: A set of 83 intermediate-difficulty problems from the American Mathematics Competitions, primarily in multiple-choice format (Li et al., 2024). (3) **MATH500**: A randomly sampled subset of 500 problems from the MATH dataset, spanning algebra, geometry, and number theory (Hendrycks et al., 2021). (4) **Min-**

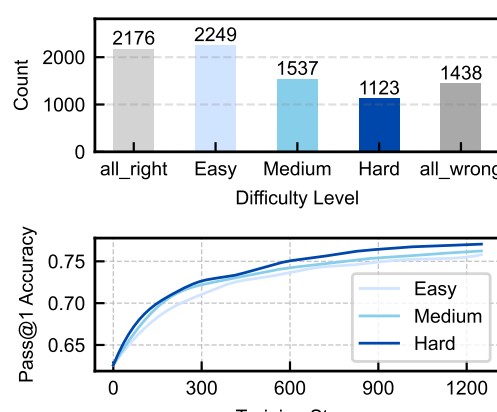

Figure 3: **Analysis of Problem Difficulty Distribution and Training Dynamics.** (a) Distribution of problems across difficulty levels. (b) Training curves of Pass@1 accuracy.

**ervaMath**: A benchmark of 272 multi-step reasoning problems (Lewkowycz et al., 2022). (5) **OlympiadBench**: A challenging suite of 675 high-difficulty mathematics problems (He et al., 2024).

**Models and Baselines.** Our experiments employ several models from the Llama and Qwen families, including `Llama-3.2-3B-Instruct` (Dubey et al., 2024), `Qwen2.5-Math-1.5B` and `Qwen2.5-Math-7B` (Yang et al., 2024) . We compare EXPO against several strong baselines: (1) **SimpleRL-Zero**: a replicate of the DeepSeek-R1 training on small models with limited data (Zeng et al., 2025b). (2) **OpenReasoner-Zero**: an open source implementation of large-scale reasoning-oriented RL training (Hu et al., 2025). (3) **PRIME-Zero**: process reinforcement through implicit rewards (Cui et al., 2025). (4) **Oat-Zero**: an unbiased optimization method that improves token efficiency while maintaining reasoning performance (Dr.GRPO; Liu et al., 2025b) . (5) **DAPO**: a decoupled clip and dynamic sampling policy optimization algorithm (Yu et al., 2025).

**Evaluation Metrics.** Following established practice (Liu et al., 2025b; Zeng et al., 2025a), our primary metric is **Pass@1** (Chen et al., 2021). Pass@k measures whether at least one of $k$ independently generated solutions is correct. We focus on the more strict Pass@1 setting, which evaluates the accuracy of a single generated response and serves as a robust indicator of model reliability.

**Implementation Details.** We conduct reinforcement learning training using the `verl` framework (Sheng et al., 2024). We set the clipping threshold to $\varepsilon = 0.2$, and during training, we sample 16 rollouts per prompt at a temperature of 1.0, with a maximum response length of 2048 tokens. The global batch size is 16, with a per-GPU mini-batch size of 4 and a learning rate of $1 \times 10^{-6}$. For inference, we use the vLLM library (Kwon et al., 2023), setting temperature to 0.0 and top-p to 1.0. To ensure rigorous evaluation on mathematical problems, we incorporate verification functions from Math-Verify. All experiments are conducted on a cluster of 1 compute node, equipped with 4 NVIDIA A40 40GB GPUs.

### 5.2 MAIN RESULTS

We present the main results of EXPO versus strong baselines on five challenging math reasoning benchmarks in Table 1. EXPO achieves the best overall performance, reaching 52.1% average accuracy using only 1,000 training samples, demonstrating its efficiency and strong ability to improve

complex reasoning. This is a +17.1% gain over the base model and +9.1% over its instruction-tuned version, showing the clear benefit of our method. Compared to existing RLVR methods, EXPO consistently performs better. It outperforms the strongest prior baseline, Oat-Zero, by 2.0% on average. While some methods excel on specific tasks, for example, Oat-Zero on AIME24 and OpenReasoner-Zero on Math500 and OlympiadBench, EXPO delivers more balanced results: it ranks first on MinervaMath and AMC23, and second on AIME24 and Math500. The most convincing evidence of EXPO's effectiveness comes from comparisons with our own reimplementations. EXPO beats standard GRPO by +5.4% on average and also surpasses recent variants like Dr.GRPO by +5.3% and DAPO by +4.6%. These consistent gains confirm that EXPO's design enables more effective learning and delivers better final performance.

Table 1: **Performance Comparison of Various Baselines on Multiple Benchmarks.** Previous RLVR methods and our implementation are based on Qwen2.5-Math-7B. **Avg.** indicates mean accuracy across all test datasets. Top results are in **bold**, and runner-up results are underlined. Performance improvements ($\Delta$) are relative to each baseline method.

| Algorithm | AIME24 | Math500 | OlympiadBench | MinervaMath | AMC23 | Avg. | $\Delta$ |
|---|---|---|---|---|---|---|---|
| Qwen2.5-Math-7B | 14.7 | 64.0 | 30.7 | 27.2 | 38.6 | 35.0 | + 17.1 |
| Qwen2.5-Math-7B-Instruct | 12.5 | 80.4 | 41.0 | 32.7 | 48.5 | 43.0 | + 9.1 |
| *Previous RLVR methods* | | | | | | | |
| SimpleRL-Zero | 27.0 | 76.0 | 34.7 | 25.0 | 54.9 | 43.5 | + 8.6 |
| OpenReasoner-Zero | 16.5 | **82.4** | **47.1** | 33.1 | 52.1 | 46.2 | + 5.9 |
| PRIME-Zero | 17.0 | 81.4 | 40.3 | 39.0 | 54.0 | 46.3 | + 5.8 |
| Oat-Zero | **33.4** | 78.0 | 43.4 | 34.6 | 61.2 | 50.1 | + 2.0 |
| *Our Implementation* | | | | | | | |
| GRPO | 14.8 | 80.0 | 42.1 | 41.2 | 55.4 | 46.7 | + 5.4 |
| Dr.GRPO | 16.6 | 81.2 | 43.4 | 44.5 | 48.2 | 46.8 | + 5.3 |
| DAPO | 13.3 | 79.6 | 39.3 | 43 | 62.5 | 47.5 | + 4.6 |
| EXPO (ours) | 30.0 | 81.8 | 41.3 | **44.9** | **62.5** | **52.1** | |

## 5.3 ANALYSIS

### 5.3.1 TRAINING DYNAMICS

We compare the training dynamics of EXPO and DAPO across three key metrics: policy entropy, reward score, and response length. As shown in the left panel of Figure 4, EXPO maintains higher entropy throughout training compared to DAPO, indicating a more exploratory behavior and sustained diversity in generated responses. This aligns with EXPO's design goal of amplifying rare, high-reward trajectories, preventing premature convergence to narrow policy regions. The middle panel reveals that EXPO achieves a consistently higher reward score, demonstrating its effectiveness in optimizing for quality while preserving exploration. Notably, the reward improvement is not accompanied by a reduction in response length (right panel), where both methods exhibit similar trends, initially decreasing before stabilizing around 700-800 tokens. However, EXPO maintains slightly longer responses on average, suggesting it preserves expressiveness without sacrificing coherence or reward. Together, these dynamics confirm that EXPO strikes a better balance between exploration and exploitation, enabling robust learning from rare but valuable samples.

### 5.3.2 IMPACT OF LLM BACKBONE

To verify that the advantages of EXPO are not limited to a single model architecture, we extended our evaluation to different LLM backbones. As shown in Table 2, we conducted experiments on smaller Qwen2.5-Math-1.5B and Llama-3.2-3B-Instruct models. On the Qwen2.5-Math-1.5B, EXPO once again achieves the highest average accuracy of 43.9%, delivering a substantial improvement of +3.5% over the standard GRPO baseline and +1.8% over the strong competitor, Dr. GRPO. More importantly, we observe a similar trend on the Llama-3.2 model. Despite this model having a different architecture and a lower initial performance, EXPO still emerges as the most effective algorithm with an average accuracy of 24.8%. This represents a clear gain of +1.6% over GRPO and even larger gains over Dr. GRPO by +4.0% and DAPO +3.6 %. These consistent results across two

distinct model families robustly demonstrate that the performance improvements offered by EXPO are a general property of our algorithm and not dependent on a specific model architecture.

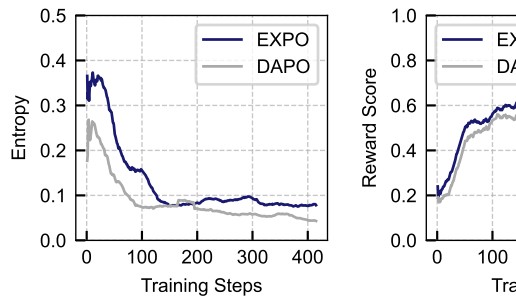 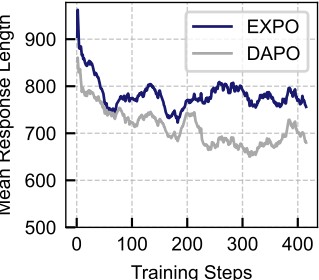

Figure 4: **Training Dynamics of EXPO.** Left: Policy entropy, showing EXPO maintains higher exploration throughout training. Middle: Reward score, indicating EXPO achieves consistently higher performance. Right: Mean response length, revealing both methods stabilize at similar lengths, with EXPO slightly preserving longer outputs.

Table 2: **Performance Comparison under Different LLM Backbones.** Qwen2.5-Math-1.5B and Llama-3.2-3B-Instruct are evaluated. The results highlight the impact of both model architecture and optimization strategy, with EXPO achieving the highest average scores across both LLM backbones, demonstrating its consistent effectiveness in enhancing reasoning performance.

| Algorithm | AIME24 | Math500 | OlympiadBench | MinervaMath | AMC23 | Avg. | $\Delta$ |
|---|---|---|---|---|---|---|---|
| Qwen2.5-Math-1.5B | 10.0 | 62.2 | 29.2 | 16.2 | 42.5 | 32.0 | +11.9 |
| + GRPO | 13.3 | 73.2 | 32.7 | 30.1 | 52.5 | 40.4 | +3.5 |
| + Dr. GRPO | 20.0 | 74.2 | 37.6 | 25.7 | 53.0 | 42.1 | +1.8 |
| + DAPO | 13.3 | 71.8 | 32.3 | 29.4 | 57.5 | 40.9 | +3.0 |
| + EXPO | 23.3 | 71.6 | 34.1 | 30.5 | 60.0 | 43.9 | |
| Llama-3.2-3B-Instruct | 6.7 | 41.0 | 12.1 | 17.3 | 15.0 | 18.4 | +6.4 |
| + GRPO | 6.7 | 44.8 | 17.2 | 22.1 | 25.0 | 23.2 | +1.6 |
| + Dr. GRPO | 6.7 | 50.0 | 14.7 | 14.3 | 18.1 | 20.8 | +4.0 |
| + DAPO | 6.7 | 45.2 | 14.2 | 19.9 | 20.0 | 21.2 | +3.6 |
| + EXPO | 6.7 | 50.6 | 16.3 | 22.8 | 27.5 | 24.8 | |

### 5.3.3 IMPACT OF $\gamma$ COEFFICIENT

The hyperparameter $\gamma$ controls the sensitivity of the dynamic reweighting factor $\alpha_i$ to response rarity. We conduct an ablation study to evaluate its impact on performance and training dynamics. As shown in left panel of Figure 5, Pass@1 accuracy peaks at $\gamma = 1.0$, with a drop for both lower and higher values. This indicates that moderate amplification is optimal: too weak ($\gamma < 1.0$) fails to boost rare high-reward responses, while excessive amplification ($\gamma > 1.0$) may overemphasize noise or unstable signals. The middle panel shows that larger $\gamma$ values accelerate early convergence, as seen in steeper initial curves for $\gamma = 1.5$ and 2.0. However, these models tend to plateau earlier, suggesting that aggressive amplification may lead to premature stabilization.

Table 3: **Average Number of Correct Solutions across Mathematical Benchmarks.** Higher values indicate greater diversity in valid reasoning solutions. All results are based on the Qwen2.5-Math-7B.

| Algorithm | DAPO | EXPO |
|---|---|---|
| AIME24 | 25.8 | 27.2 |
| Math500 | 98.5 | 100.8 |
| OlympiadBench | 78.7 | 79.9 |
| MinervaMath | 41.4 | 42.3 |
| AMC23 | 48.4 | 50.0 |

Finally, the right panel reveals how $\gamma$ shapes the evolution of dynamic weights. Higher $\gamma$ leads to stronger initial amplification and faster decay of weight magnitude, indicating more aggressive focus on rare events early in training, followed by rapid deemphasis. Together, these results confirm that $\gamma = 1.0$ strikes the best balance between exploration

and stability, enabling EXPO to effectively learn from rare valuable responses without sacrificing training robustness.

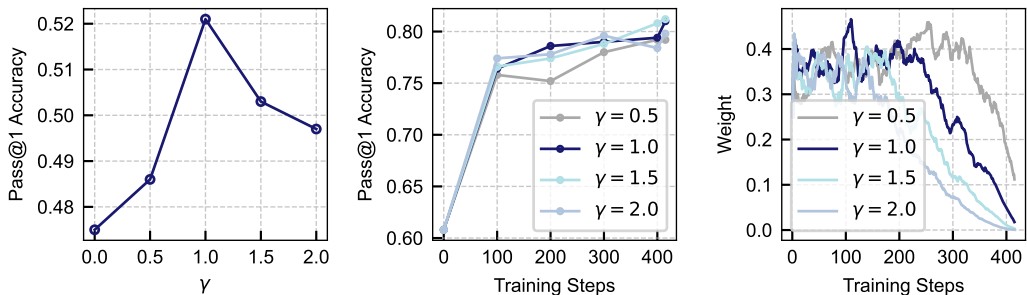

Figure 5: **Ablation Study of Reweighting Factor $\gamma$ in EXPO.** (a) Pass@1 accuracy vs. $\gamma$ : peak performance at $\gamma = 1.0$ . (b) Training curves for different $\gamma$ : higher values improve early convergence but may plateau earlier. (c) Evolution of dynamic weights: larger $\gamma$ leads to stronger amplification and faster decay of rare-response weights.

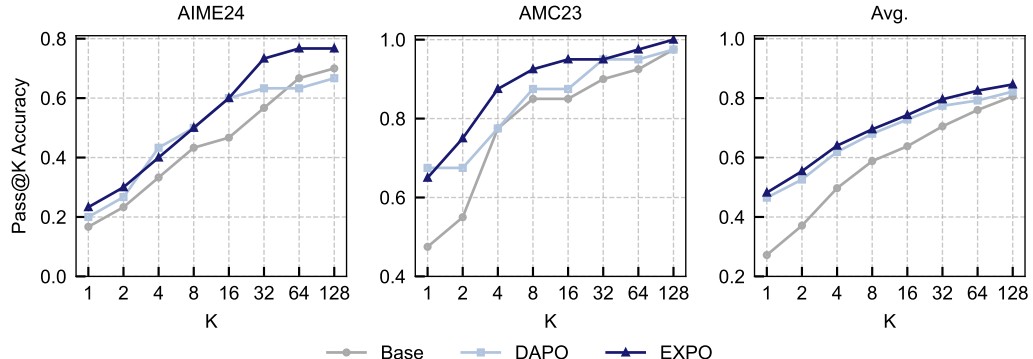

Figure 6: **Pass@K Accuracy across Mathematical Reasoning Benchmarks.** EXPO consistently outperforms both the base model and DAPO at all values of K. More results on Math500, Olympiad-Bench and MinervaMath can be found at Appendix C

### 5.3.4 DIVERSITY OF REASONING SOLUTIONS

To quantify the diversity of correct reasoning solutions generated by the model, we sample 128 solutions per problem using a high temperature $T = 1.0$ to encourage stochasticity and avoid repetitive outputs. We then count the average number of correct solutions across five mathematical benchmarks. This metric reflects the breadth of the model's reasoning, an indicator of creative and robust problem-solving. As shown in the table 3, EXPO consistently generates more diverse correct solutions than DAPO across all benchmarks, with the largest improvements on Math500 (+2.3) and AMC23 (+1.6). This is a direct consequence of EXPO's core design. By dynamically amplifying gradients from rare valuable trajectories, EXPO actively resists premature convergence to dominant or repetitive solution patterns. As a result, the model preserves and refines a wider variety of valid reasoning paths throughout training, enabling it to internalize and deploy a richer, more diverse set of problem-solving strategies at inference time.

### 5.3.5 EFFECT ON REASONING CAPACITY BOUNDARY

The core function of EXPO is to encourage the generation of diverse reasoning solutions. A natural question is: does this diversity help improve reasoning capability boundary? To investigate this, we adopt the pass@k metric following Yue et al. (2025), which measures whether a model can generate

at least one correct solution in $k$ independent attempts. This provides an upper-bound estimate of the model's latent reasoning capacity. As shown in Figure 6, we observe a clear divergence in trends between small and large $k$. At small $k$, RLVR-trained models outperform the base model, consistent with the well-known finding that RLVR improves single-sample correctness. However, as $k$ increases, the base model steadily closes the gap. For example, on AIME24, DAPO leads the base model by +3.3% at $k = 1$, but falls behind by -3.3% at $k = 128$, suggesting that DAPO's gains come from concentrating probability mass on known correct paths, rather than discovering new ones. In contrast, EXPO maintains a consistent performance advantage over the base model even at $k = 128$, achieving a final average accuracy of 84.6% versus 80.6%, a +4.0% absolute gain. This demonstrates that EXPO not only improves single-attempt accuracy but also genuinely expands the model's reasoning boundary by preserving and reinforcing diverse, high-reward solution trajectories.

## 6 Related Work

**Mathematical Reasoning with LLMs**   Mathematical reasoning is a gold standard for evaluating LLMs (Zhang et al., 2025), requiring symbolic abstraction, logical consistency, and multi-step deduction, cognitive traits central to science and engineering. Research mainly splits into two paradigms. *Formal reasoning*, based on systems like Lean or Coq (Zheng et al., 2022; Azerbayev et al., 2023; Xin et al., 2025), ensures correctness via machine-checkable proofs, ideal for theorem proving. *Informal reasoning*, using natural language or code without formal guarantees, better mirrors human problem solving: flexible, heuristic, and often tool-free (Sun & Zhang, 2025; Singh et al., 2025). It excels in tasks like word problems and symbolic computation, where plausible, high-quality outputs matter more than formal proof. We adopt informal reasoning, as it matches real-world settings where formal systems are unavailable. In such settings, models must learn from sparse rewards and discover rare, high-value reasoning paths. This is the core challenge EXPO addresses by amplifying signals from low-probability, high-reward responses.

**Policy Optimization for LLMs**   Reinforcement learning has significantly improved LLM reasoning, as shown in models like OpenAI-o3, DeepSeek-R1 and Kimi-K2 (OpenAI, 2025; DeepSeek-AI et al., 2025; Bai et al., 2025). Progress largely builds on verifiable rewards (Zeng et al., 2025b; Hu et al., 2025; Cui et al., 2025), which offer reliable training signals. Follow-up work uses test-time adaptation (Muennighoff et al., 2025; Zuo et al., 2025) and structured prompting (Wang et al., 2023; Sun & Zhang, 2025) to boost performance within existing limits, while newer RL methods (Shao et al., 2024a; Liu et al., 2025a; Yu et al., 2025) refine objectives for reasoning, yet remain mostly on-policy, amplifying known behaviors instead of discovering new ones. Recent work (Zhao et al., 2025; Yue et al., 2025) identifies a key issue: on-policy learning rarely explores beyond current behavior, favoring safe, frequent outputs and optimize within model boundaries rather than expanding reasoning horizons. EXPO tackles this by dynamically reweighting gradients to amplify signals from low-probability, high-reward responses. This enables learning from the model's most valuable, previously ignored outputs, preserving reasoning diversity and preventing model collapse into narrow solution modes, thereby improving overall performance.

## 7 Conclusion

We identify a critical limitation in standard policy gradient methods for LLM alignment: their tendency to suppress low-probability, high-reward responses, a bias that narrows reasoning scope, collapses diversity, and stifles creative problem-solving. To address this, we propose Exploration-Enhanced Policy Optimization, a lightweight algorithm that dynamically reweights policy gradients to amplify learning signals from rare but valuable outputs while penalizing frequent mistakes. By reshaping the optimization objective around response rarity and reward, EXPO enables models to escape local optima, sustain exploration, and internalize diverse reasoning strategies. Extensive experiments across multiple mathematical benchmarks confirm that EXPO consistently outperforms strong baselines like DAPO and GRPO, achieving higher accuracy, greater solution diversity, and more stable training, demonstrating that explicitly valuing happy accidents is not just beneficial, but essential for unlocking the full reasoning potential of large language models.

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

## A TRAINING DATA

We build our training set from MATH dataset problems at difficulty levels 3–5. For each problem, we generate four responses using Qwen2.5-Math-7B and classify it into one of five difficulty tiers based on correctness: all-right: 4 correct, Easy: 3 correct, Medium: 2 correct, Hard: 1 correct, all-wrong: 0 correct. We exclude all-right and all-wrong problems, as they offer little learning signal, being either too easy or too hard. This leaves Easy, Medium, and Hard problems to study how data difficulty affects RL fine-tuning. As shown in Figure 7, training on Hard problems yields faster convergence and sustained gains. From the 1,123 problems in the Hard tier, we randomly sample 1,000 for our final training set, balancing challenge and tractability to maximize learning signal.

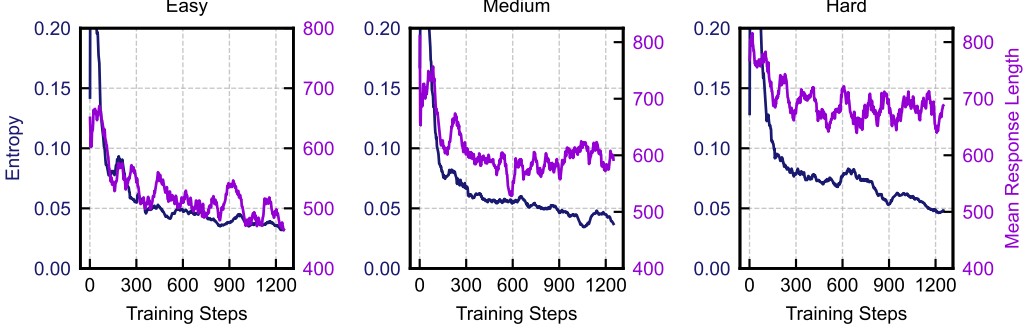

Figure 7: **Training Dynamics by Difficulty Level.** Entropy (blue) and mean response length (purple) across training steps for Easy, Medium, and Hard problems. Hard problems show sustained variability.

## B CODE IMPLEMENTATION

It is easy to implement EXPO based on open-source RL framework. For example, we show the minimum viable implementation of EXPO that only modifies a few line of DAPO loss in verl.

```python
def compute_policy_loss(advantages, sentence_logps, step_ratio, gamma
    =1.0):
    mask_adv = advantages.mean(dim=1) < 0
    sentence_logps_detach = torch.clamp(torch.exp(sentence_logps.detach()
        ), 0, 1)
    sentence_logps_detach[mask_adv] = 1 - sentence_logps_detach[mask_adv]
    sentence_logps_detach = torch.clamp(sentence_logps_detach, step_ratio
        , 1)
    alpha = 1 - sentence_logps_detach
    alpha = alpha.pow(gamma)
    alpha = torch.clamp(alpha, 0, 0.5)
    return alpha
```

Listing 1: Function `compute_policy_loss` for adaptive response weighting.

## C   MORE EVALUATION RESULTS

We evaluate Qwen2.5-Math-7B under three settings: (1) Base, (2) DAPO, and (3) EXPO. Results are reported across five mathematical benchmarks using pass@K with $K \in \{1, 2, 4, 8, 16, 32, 64, 128\}$, sampling temperature T=1.0 , and correctness judged by any correct sample. As shown in below tables, EXPO consistently outperforms both Base and DAPO at all K values. With Qwen2.5-math-7B, The gap is largest at K=1, highlighting EXPO's strength in single-sample accuracy, crucial for real-world use. At K=128 , EXPO achieves 84.6% average, surpassing DAPO and Base. These results confirm that EXPO improves not only accuracy but also the diversity and reasoning capability boundary.

Table 4: Pass@K performance of Llama-3.2-3B-Instruct without finetuning across multiple mathematical reasoning benchmarks. K denotes the number of sampled responses per problem, with success measured if any response is correct. All evaluations use identical prompting, decoding, and temperature settings for fair comparison.

| K | AIME24 | Math500 | OlympiadBench | Minerva | AMC23 | Avg. |
|---|--------|---------|---------------|---------|-------|------|
| 1 | 0.0 | 25.0 | 5.5 | 7.7 | 2.5 | 8.1 |
| 2 | 3.3 | 40.2 | 9.3 | 11.0 | 22.5 | 17.3 |
| 4 | 3.3 | 50.6 | 15.6 | 17.3 | 25.0 | 22.4 |
| 8 | 6.7 | 60.6 | 21.2 | 24.6 | 35.0 | 29.6 |
| 16 | 6.7 | 71.2 | 29.3 | 30.9 | 60.0 | 39.6 |
| 32 | 6.7 | 77.8 | 38.7 | 37.5 | 70.0 | 46.1 |
| 64 | 16.7 | 86.0 | 46.4 | 41.9 | 85.0 | 55.2 |
| 128 | 26.7 | 88.4 | 53.5 | 48.2 | 92.5 | 61.9 |

Table 5: Pass@K performance of Llama-3.2-3B-Instruct finetuned with DAPO across multiple mathematical reasoning benchmarks.

| K | AIME24 | Math500 | OlympiadBench | Minerva | AMC23 | Avg. |
|---|--------|---------|---------------|---------|-------|------|
| 1 | 3.3 | 44.6 | 13.8 | 16.2 | 30.0 | 21.6 |
| 2 | 6.7 | 53.4 | 18.7 | 23.2 | 32.5 | 26.9 |
| 4 | 6.7 | 61.0 | 24.4 | 27.9 | 40.0 | 32.0 |
| 8 | 16.7 | 66.6 | 28.7 | 32.7 | 42.5 | 37.4 |
| 16 | 20.0 | 72.2 | 33.2 | 39.3 | 55.0 | 43.9 |
| 32 | 20.0 | 76.0 | 37.9 | 43.4 | 62.5 | 48.0 |
| 64 | 26.7 | 79.4 | 42.1 | 49.3 | 67.5 | 53.0 |
| 128 | 30.0 | 82.6 | 45.9 | 53.7 | 72.5 | 56.9 |

Table 6: Pass@K performance of Llama-3.2-3B-Instruct finetuned with EXPO across multiple mathematical reasoning benchmarks.

| K | AIME24 | Math500 | OlympiadBench | Minerva | AMC23 | Avg. |
|---|--------|---------|---------------|---------|-------|------|
| 1 | 6.7 | 44.4 | 14.5 | 16.5 | 32.5 | 22.9 |
| 2 | 10.0 | 55.4 | 19.7 | 23.5 | 35.0 | 28.7 |
| 4 | 10.0 | 63.6 | 24.7 | 28.7 | 35.0 | 32.4 |
| 8 | 13.3 | 70.0 | 29.9 | 32.4 | 50.0 | 39.1 |
| 16 | 13.3 | 74.4 | 36.0 | 38.2 | 57.5 | 43.9 |
| 32 | 16.7 | 81.2 | 40.3 | 44.5 | 65.0 | 49.5 |
| 64 | 26.7 | 85.4 | 47.0 | 49.6 | 72.5 | 56.2 |
| 128 | 33.3 | 88.4 | 51.0 | 57.0 | 85.0 | 62.9 |

Table 7: Pass@K performance of Qwen2.5-math-7B without finetuning DAPO across multiple mathematical reasoning benchmarks.

| K | AIME24 | Math500 | OlympiadBench | Minerva | AMC23 | Avg. |
|---|--------|---------|---------------|---------|-------|------|
| 1 | 16.7 | 42.4 | 18.2 | 11.0 | 47.5 | 27.2 |
| 2 | 23.3 | 58.4 | 30.2 | 18.4 | 55.0 | 37.1 |
| 4 | 33.3 | 73.2 | 39.3 | 25.0 | 77.5 | 49.7 |
| 8 | 43.3 | 84.8 | 47.9 | 33.1 | 85.0 | 58.8 |
| 16 | 46.7 | 90.0 | 55.6 | 41.5 | 85.0 | 63.8 |
| 32 | 56.7 | 93.6 | 63.6 | 48.5 | 90.0 | 70.5 |
| 64 | 66.7 | 95.0 | 69.3 | 56.6 | 92.5 | 76.0 |
| 128 | 70.0 | 96.6 | 75.6 | 63.2 | 97.5 | 80.6 |

Table 8: Pass@K performance of Qwen2.5-math-7B finetuned with DAPO across multiple mathematical reasoning benchmarks.

| K | AIME24 | Math500 | OlympiadBench | Minerva | AMC23 | Avg. |
|---|--------|---------|---------------|---------|-------|------|
| 1 | 20.0 | 76.2 | 37.8 | 30.9 | 67.5 | 46.5 |
| 2 | 26.7 | 83.4 | 45.2 | 40.4 | 67.5 | 52.6 |
| 4 | 43.3 | 88.4 | 51.9 | 48.2 | 77.5 | 61.9 |
| 8 | 50.0 | 90.6 | 58.8 | 52.9 | 87.5 | 68.0 |
| 16 | 60.0 | 93.0 | 64.7 | 58.8 | 87.5 | 72.8 |
| 32 | 63.3 | 94.8 | 69.2 | 64.7 | 95.0 | 77.4 |
| 64 | 63.3 | 96.2 | 72.6 | 68.8 | 95.0 | 79.2 |
| 128 | 66.7 | 98.0 | 76.3 | 72.4 | 97.5 | 82.2 |

Table 9: Pass@K performance of Qwen2.5-math-7B finetuned with EXPO across multiple mathematical reasoning benchmarks.

| K | AIME24 | Math500 | OlympiadBench | Minerva | AMC23 | Avg. |
|---|--------|---------|---------------|---------|-------|------|
| 1 | 23.3 | 79.2 | 39.6 | 33.8 | 65.0 | 48.2 |
| 2 | 30.0 | 85.6 | 47.4 | 39.0 | 75.0 | 55.4 |
| 4 | 40.0 | 89.6 | 54.7 | 48.2 | 87.5 | 64.0 |
| 8 | 50.0 | 92.4 | 58.8 | 54.0 | 92.5 | 69.5 |
| 16 | 60.0 | 93.8 | 64.3 | 58.5 | 95.0 | 74.3 |
| 32 | 73.3 | 95.0 | 69.5 | 65.1 | 95.0 | 79.6 |
| 64 | 76.7 | 96.6 | 72.9 | 68.8 | 97.5 | 82.5 |
| 128 | 76.7 | 97.6 | 76.4 | 72.1 | 100.0 | 84.6 |

# D CASE STUDY

## D.1 COMPARISON BETWEEN EXPO AND BASELINES

---

**Question**: Find the greatest common divisor of 3339, 2961, and 1491. (Correct answer is $\boxed{21}$).

---

**GRPO**:
To find the greatest common divisor (GCD) of the numbers 3339, 2961, and 1491, we can use the Euclidean algorithm. The Euclidean algorithm is an efficient method for computing the greatest common divisor of two numbers. We can extend this method to more than two numbers by iteratively applying it to pairs of numbers. The steps are as follows: 1. First, find the GCD of the first two numbers, 3339 and 2961. 2. Then, find the GCD of the result from step 1 with the third number, 1491. Let's start with the first step: finding the GCD of 3339 and 2961 using the Euclidean algorithm. The Euclidean algorithm states that $GCD(a, b) = GCD(b, a \mod b)$ and continues until the remainder is 0. The last non-zero remainder is the GCD of the two numbers. We will apply this iteratively, the GCD of 3339 and 2961 is 126. Now, we need to find the GCD of 126 and 1491. The GCD of 126 and 1491 is 9. Therefore, the GCD of 3339, 2961, and 1491 is 9. The final answer is $\boxed{9}$.

---

**DR.GRPO**:
To find the greatest common divisor (GCD) of the numbers 3339, 2961, and 1491, we can use the Euclidean algorithm. The Euclidean algorithm finds the GCD of two numbers by repeatedly applying the division algorithm: the GCD of $a$ and $b$ is the same as the GCD of $b$ and the remainder of $a$ divided by $b$. First, we can find the GCD of 3339 and 2961 using the Euclidean algorithm. Then, we can use that GCD to find the GCD with the third number, 1491. Here's the step-by-step approach: 1. Calculate the GCD of 3339 and 2961. 2. Use the result to calculate the GCD with 1491. Let's start by calculating the GCD of 3339 and 2961. We use the Euclidean algorithm, the GCD of 3339 and 2961 is 1. Since the GCD of 3339 and 2961 is 1, the GCD of 1 and any other number is 1. So, the GCD of 1 and 1491 is 1. The final answer is $\boxed{1}$.

---

**DAPO**:
To find the greatest common divisor (GCD) of 3339, 2961, and 1491, we will first find the GCD of 3339 and 2961 using the Euclidean algorithm, and then we will find the GCD of the result with 1491. The Euclidean algorithm for finding the GCD of two numbers $a$ and $b$ (where $a > b$) is: 1. Divide $a$ by $b$ to get the remainder $r$. 2. Replace $a$ with $b$ and $b$ with $r$. 3. Repeat the process until the remainder is 0. The last non-zero remainder is the GCD of $a$ and $b$. Let's first find the GCD of 3339 and 2961. We will then find the GCD of the result with 1491. The greatest common divisor (GCD) of 3339, 2961, and 1491 is 213. The final answer is $\boxed{3}$.

---

**EXPO**:
To find the greatest common divisor (GCD) of 3339, 2961, and 1491, we can use the Euclidean algorithm. The Euclidean algorithm is an efficient method for computing the greatest common divisor (GCD) of two numbers. We can extend this method to more than two numbers by iteratively applying the algorithm. The steps to find the GCD of multiple numbers are as follows: 1. Find the GCD of the first two numbers. 2. Use the result to find the GCD with the next number. 3. Repeat until you have the GCD of all the numbers. We start by finding the GCD of 3339 and 2961, and then we find the GCD of that result with 1491. Let's do this step-by-step. First, we find the GCD of 3339 and 2961, the GCD of 3339 and 2961 is the last non-zero remainder, which is 3. Next, we find the GCD of 3 and 1491: 1491 = 3 * 497 + 0 The GCD of 3 and 1491 is 3. The final answer is $\boxed{21}$.

---

## D.2 DIFFERENT SOLUTIONS FOR THE SAME PROBLEM

---

**Question**:

You have 5 shirts, 6 pairs of pants, and 8 hats. How many outfits can you make consisting of one shirt, one pair of pants, and one hat?

(Correct answer is $\boxed{240}$).

**EXPO (Solution 1):**

To determine the total number of different outfits that can be made, we need to calculate the product of the number of choices for each item of clothing. Specifically, there are 5 choices for a shirt, 6 choices for a pair of pants, and 8 choices for a hat. The total number of different outfits can be calculated as follows: Total outfits = Number of shirts × Number of pairs of pants × Number of hats = $5 \times 6 \times 8$ We can calculate this product step by step. First, we calculate the product of the number of shirts and pairs of pants: $5 \times 6 = 30$ Then, we multiply the result by the number of hats: $30 \times 8 = 240$ So, the total number of different outfits that can be made is 240. The final answer is $\boxed{240}$.

**EXPO (Solution 2):**

We have 5 shirts, 6 pairs of pants, and 8 hats. To find the total number of possible outfits consisting of one shirt, one pair of pants, and one hat, we can use the principle of counting combinations. The total number of possible outfits is the product of the number of choices for each item. So, we need to calculate: Total outfits = (Number of shirts) × (Number of pairs of pants) × (Number of hats) Given: Number of shirts = 5, Number of pairs of pants = 6, Number of hats = 8. So, the total number of outfits is: Total outfits = $5 \times 6 \times 8$. The total number of possible outfits consisting of one shirt, one pair of pants, and one hat is 240. The final answer is $\boxed{240}$.

**EXPO (Solution 3):**

You have 5 shirts, 6 pairs of pants, and 8 hats. To determine how many different outfits can be made consisting of one shirt, one pair of pants, and one hat, we can use the basic principle of counting. According to the principle of counting, if there are $m$ ways to do one thing and $n$ ways to do another, then there are $m \times n$ ways to do both. Extending this to three choices, if there are $a$ ways to do the first thing, $b$ ways to do the second, and $c$ ways to do the third, then there are $a \times b \times c$ ways to do all three. In this problem: There are 5 ways to choose a shirt. There are 6 ways to choose a pair of pants. There are 8 ways to choose a hat. Therefore, the total number of different outfits is $5 \times 6 \times 8$. The total number of different outfits that can be made consisting of one shirt, one pair of pants, and one hat is 240. The final answer is $\boxed{240}$.

# E    LLM USAGE STATEMENT

LLMs were used in the preparation of this paper solely for grammatical correction and language polishing, and they help improve the clarity and fluency of the writing. All intellectual contributions, including concept development, algorithm design, experimentation, and interpretation of results, are entirely the work of the authors.

