# OpenReview forum: "Improving Policy Optimization via Enhanced Exploration"
_ICLR.cc/2026/Conference — ICLR 2026 Conference Withdrawn Submission_

### Official Review · Reviewer_2Hs1 · 2025-10-22

**Soundness:** 2
**Presentation:** 2
**Contribution:** 2
**Rating:** 4
**Confidence:** 4

**Summary:**

This paper identifies a key limitation in standard RL methods used for aligning LLMs, which the authors term "statistical short-sightedness." They argue that existing policy gradient algorithms are biased towards high-probability responses, causing them to overlook and suppress rare, yet valuable, reasoning paths. To address this, they propose an algorithm that modifies the advantage function by dynamically reweighting it based on the generation probability of a response. This mechanism amplifies the learning signal for low-probability, high-reward outputs and more heavily penalizes high-probability, low-reward outputs. The authors demonstrate through extensive experiments on mathematical reasoning benchmarks that EXPO consistently outperforms the baseline algorithms.

**Strengths:**

1. The paper tackles a critical problem in the RLHF of LLMs that the tendency of policy optimization to reduce entropy and converge on a narrow set of "safe" solutions. The work focuses on enhancing exploration to escape local optima. And it makes clear derivation from the base algorithm of DAPO. While the idea of reweighting samples is not entirely new in RL, its specific formulation is quite novel.

2. This paper including a lot of experiments on different models and downstream tasks to empirically validate the efficiency of the proposed method. And this paper also includes ablation studies on $\gamma$ and insights on training dynamics, providing a comprehensive view of the improvement of the proposed approach.

**Weaknesses:**

1. The proposed method is a reweighting scheme applied directly on top of the DAPO framework. While effective, it feels more like an incremental improvement than a fundamentally new algorithm. The specific functional form for the dynamic weight, $\alpha_{i}=clip((1-clip(\tilde{p}\_{i},\delta,1))^\gamma,0,\alpha_{\max})$, is presented without strong theoretical justification and feels somewhat heuristic.

2. The novel approach introduces several new hyperparameters that appear crucial for its stability and performance including $\gamma$, $\alpha_{\max}$, and $\delta$. The paper includes ablation study experiments for $\gamma$, but the sensitivity to $\alpha_{max}$ and the design of the $\delta$ schedule are not explored. This added complexity may make the algorithm less practical.

Typos:

1. Table 1: 43->43.0

**Questions:**

1. Could you explain more on Figure 5 (a) and (b)? It shows that both figures have y-label "Pass@1 Accuracy". But the results can not match for certain $\gamma$.
2. Could you provide a deeper justification for the specific mathematical form of $\alpha_i$? Were alternative functions for reweighting based on probability explored, and if so, how did they perform?
3. For low-reward responses ($A_i < 0$), the proposed method is designed to more heavily penalize frequent mistakes. However, it seems distinct from the primary goal of enhancing exploration of rare positive discoveries. What was the reason for this design choice, and have you tested an alternative approach where you only amplify rare positive rewards while applying a standard penalty to all negative ones?

---

### Official Review · Reviewer_anbT · 2025-10-26

**Soundness:** 2
**Presentation:** 3
**Contribution:** 2
**Rating:** 2
**Confidence:** 4

**Summary:**

The authors propose an exploration-enhanced policy optimization (EXPO) algorithm which is aimed to overcoming the limitations of policy gradient algorithms, i.e. GRPO and DAPO can under-sample rare valuable responses because their gradients are expectation-weighted by the current policy. While DAPO improves gradient estimation, it does not solve the underrepresentation bias. The proposed EXPO algorithm uses a dynamic advantage weighting scheme to ensure gradients from outputs that are high-reward but low-probability under the current policy are taken into account. The authors evaluate the proposed algorithm on multiple mathematical reasoning benchmarks where it outperforms state-of-the-art baselines across model scales.

**Strengths:**

1. The authors address a fundamental weakness in policy-gradient algorithms for training LLMs.

2. The experimental testbed includes a number of ablation studies to compare with DAPO.

**Weaknesses:**

1. The fundamental issue raised by the authors (rare but valuable responses are undersampled) seems to me to remain unaddressed because advantage estimation is often biased or mis-signed for low-probability responses. When you oversample rare responses, you also oversample regions where the advantage estimates have the highest variance (since these responses were rarely observed or labeled).

**Questions:**

1. The proposed algorithm introduces several additional hyper parameters ($\delta, \gamma, \alpha_max$). Can the authors elaborate why their testbed is a fair comparison with DAPO (with lower number of hyper-parameters) ?

---

### Official Review · Reviewer_BMqq · 2025-10-28

**Soundness:** 2
**Presentation:** 3
**Contribution:** 3
**Rating:** 4
**Confidence:** 3

**Summary:**

Reinforcement learning has been considered as one of the standard approaches to align LLMs to reasoning tasks. However, it often favors those high-probability responses due to the loss design. In this work, the authors propose **Exploration-Enhanced Policy Optimization (EXPO)**, an easy-to-use method that rewards the correct but low-probability responses while penalizing frequently occurring mistakes. Experimental results indicate that EXPO achieves improved overall performance compared to previous baselines.

**Strengths:**

- The proposed method is simple and easy to use while showing improvement in performance over several baselines.
- It covers a wide range of experiments and analyses that provide substantial support for the proposed method.

**Weaknesses:**

- While the overall performance is notable, the paper does not provide a clear justification for the specific design of the training dataset. It would be more convincing if the authors conducted an experiment on a uniformly sampled subset or offered an explanation for their choice. In addition, although the authors state that the **Hard** problems converge faster, they do not provide any intuition or analysis to explain this phenomenon.
- The paper lacks an analysis of the cases where the proposed method underperforms compared to other baselines on certain benchmarks.
- The step size of $\gamma$ in the left figure of Figure 5 is too large, making it difficult to determine whether the method is sensitive to hyperparameters.

**Questions:**

- What is the benchmark for the Pass@1 accuracy shown in the left figure of Figure 5? Would the proposed method underperform compared to other baselines if $\gamma$ is not properly tuned?
- The authors mention that the performance gap between EXPO and DAPO is largest at $K=1$, which seems counterintuitive since EXPO should generate more diverse responses, while DAPO is more centralized. Therefore, the gap would be expected to increase as $K$ grows. Is there any intuition or explanation behind this phenomenon?
- Does the distribution shift shown in Figure 1 occur only in the training dataset, or does it also appear in the testing data?

---

### Official Review · Reviewer_JuBR · 2025-10-31

**Soundness:** 2
**Presentation:** 3
**Contribution:** 2
**Rating:** 4
**Confidence:** 4

**Summary:**

The paper proposes a reinforcement learning framework that addresses GRPO's limited exploration and training instability through three components: additional rollouts for targeted exploration on difficult prompts, online filtering to remove low-quality samples, and experience replay to amplify rare high-quality trajectories.

**Strengths:**

- **Addresses a relevant problem**. The paper identifies that GRPO suffers from limited exploration and training instability on complex reasoning tasks, supported by empirical observations on failed training cases.
- **Simple and plug-and-play design**. EFRame's three components (additional rollout, filtering, replay) can be seamlessly integrated into existing GRPO-based pipelines with minimal modifications to the training framework.

**Weaknesses:**

- The experimental benchmarks are restricted to recent RLVR methods tailored to LLMs, with no direct evaluation against classic exploration RL methods adapted for LLM settings.
- While the ablation on the $\gamma$ coefficient is conducted, other hyperparameters (such as the weight bounds, and progressive adjustment schedule) are not fully explored.
- While the benchmarks are broad within math reasoning, all experiments are on mathematical reasoning datasets , with no evidence provided as to whether EXPO offers similar benefits on other domains where rare but high-reward trajectories exist.
- While the paper repeatedly asserts that EXPO "generalizes" to various backbones and scales, the only two model families tested are Qwen2.5-Math and Llama, and there are no experiments on models larger than 7B.

**Questions:**

- **Limited methodological novelty**.  EFRame combines three well-established RL techniques: additional rollouts, sample filtering, and experience replay. Several concurrent works (e.g., RePO[1], DOTS[2], ReMix[3]) employ similar filtering and replay strategies for GRPO improvement. The contribution appears incremental rather than introducing fundamentally new algorithmic insights.
- **Limited baseline comparisons**. The experimental benchmarks are restricted to recent RLVR methods (GRPO, DAPO) tailored to LLMs, with no direct evaluation against other off-policy RL methods or classic exploration techniques adapted for LLM settings.
- **Insufficient hyperparameter analysis**. While ablations on temperature ($t_a$) and replay buffer size ($R_s$) are conducted, other critical hyperparameters—such as the additional rollout size ($G_2$), filtering thresholds, and replay frequency—are not thoroughly explored across different datasets or model scales.
- **Limited domain diversity**. All experiments focus on mathematical and geometric reasoning tasks (MATH, AIME'24, Geometry3K, MathVision, etc.). No evidence is provided as to whether EFRame offers similar benefits on other domains such as code generation, planning, or general instruction following where exploration challenges may differ.
- **Narrow model scope**. While the paper claims generalizability across backbones, experiments are limited to Qwen2.5-Math-7B and Qwen2.5-VL-7B-Instruct. There are no evaluations on larger models (e.g., 13B, 70B) or other model families (e.g., Llama, DeepSeek), limiting the understanding of scalability.

**References:**

[1] RePO: Replay-Enhanced Policy Optimization

[2] DOTS: Learning to Reason Dynamically in LLMs via Optimal Reasoning Trajectories Search

[3] Squeeze the Soaked Sponge: Efficient Off-policy Reinforcement Finetuning for Large Language Model

---

> ### Author Response · Authors · 2025-11-13
>
> Dear Reviewer JuBR,
>
> Thank you for your review. It seems that your comments may not pertain to our paper. Could you kindly update your review to reflect our submission? We will promptly address your concerns and look forward to further discussion with you.
>
> Thank you for your time and attention!

---

> > ### Comment · Reviewer_JuBR · 2025-11-13
> >
> > Dear Authors,
> >
> > Thank you for bringing this to my attention. I confirm that I have now submitted the correct review for your manuscript.
> >
> > The error occurred due to managing multiple manuscript reviews within a similar research domain. I appreciate your understanding in this matter, and I am confident that my current evaluation properly addresses the specific contributions of your work.
> >
> > I look forward to any further academic discussion regarding your research.
> >
> > Best regards,

---

### Note · Authors · 2025-11-28

**Comment:**

After careful consideration and in light of the reviewers' constructive feedback, we have decided to withdraw our submission to incorporate major revisions. We sincerely appreciate the time and effort of the reviewers and area chairs.

**Withdrawal Confirmation:**

I have read and agree with the venue's withdrawal policy on behalf of myself and my co-authors.